

# Full-length transcriptome revealed the accumulation of polyunsaturated fatty acids in developing seeds of *Plukenetia volubilis*

Yijun Fu[1,*], Kaisen Huo[1,*], Xingjie Pei[1], Chongjun Liang[1], Xinya Meng[1], Xiqiang Song[1], Jia Wang[1] and Jun Niu[1]

College of Forestry, Hainan University, Haikou, Hainan, China
[*] These authors contributed equally to this work.

## ABSTRACT

**Background**. *Plukenetia volubilis* is cultivated as a valuable oilseed crop, and its mature seeds are rich in polyunsaturated fatty acids (FAs), which are widely used in food and pharmaceutical industries. Recently, next-generation sequencing (NGS) transcriptome studies in *P. volubilis* indicated that some candidate genes were involved in oil biosynthesis. The NGS were inaccuracies in assembly of some candidate genes, leading to unknown errors in date analyses. However, single molecular real-time (SMRT) sequencing can overcome these assembled errors. Unfortunately, this technique has not been reported in *P. volubilis*.

**Methods**. The total oil content of *P. volubilis* seed (PVS) was determined using Soxhlet extraction system. The FA composition were analyzed by gas chromatography. Combining PacBio SMRT and Illumina technologies, the transcriptome analysis of developing PVS was performed. Functional annotation and differential expression were performed by BLAST software (version 2.2.26) and RSEM software (version 1.2.31), respectively. The lncRNA-targeted transcripts were predicted in developing PVS using LncTar tool.

**Results**. By Soxhlet extraction system, the oil content of superior plant-type (SPT) was 13.47% higher than that of inferior plant-type (IPT) at mature PVS. The most abundant FAs were C18:2 and C18:3, among which C18:3 content of SPT was 1.11-fold higher than that of IPT. Combined with PacBio and Illumina platform, 68,971 non-redundant genes were obtained, among which 7,823 long non-coding RNAs (lncRNAs) and 7,798 lncRNA-targeted genes were predicted. In developing seed, the expressions of 57 TFs showed a significantly positive correlation with oil contents, including WRI1-like1, LEC1-like1, and MYB44-like. Comparative analysis of expression profiles between SPT and IPT implied that orthologs of FAD3, PDCT, PDAT, and DAGT2 were possibly important for the accumulation of polyunsaturated FAs. Together, these results provide a reference for oil biosynthesis of *P. volubilis* and genetic improvement of oil plants.

Corresponding authors
Jia Wang,
wangjia9201@hainanu.edu.cn
Jun Niu, niujun@hainanu.edu.cn

## INTRODUCTION

*Plukenetia volubilis* (Euphorbiaceae family) is a perennial and woody liana, which is native to the rainforests of South America. The racemose inflorescence of *P. volubilis* has several staminate flowers situated above one or few pistillate flowers (*Goyal et al., 2022*). The ovary develops from green and fleshy to brown, woody, and dehiscent with fruit maturation (*Kodahl & Sørensen, 2021*). After sowing under suitable growing conditions, the newly planted *P. volubilis* can produce flowers and fruit capsules after 6 and 8 months. In China, *P. volubilis* is used as an under-forest economic species, and has been widely cultivated in subtropical and tropical areas, including Yunnan, Guangzhou, and Hainan province. Mature *P. volubilis* seed (PVS) has high content of oil (∼50%) that is mainly composed of palmitic acid (C16:0), stearic acid (C18:0), oleic acid (C18:1), linoleic acid (C18:2), and linolenic acid (C18:3) (*Gutiérrez, Rosada & Jiménez, 2011*; *Niu et al., 2014*). The PVS oil has the distinctive character of high content of linoleic and linolenic acid, comprising about 90% (*Chirinos et al., 2013*). It has been reported in the human that the linolenic and linoleic acid showed effectiveness in preventing coronary heart disease, hypertension, diabetes, arthritis, high cholesterol, cancer, and autoimmune diseases (*Wijendran & Hayes, 2004*). People attach economic value to PVS oil more and more in food, medicine, cosmetics and other fields (*Wang, Zhu & Kakuda, 2018*).

Glycolysis is the main pathway to generate pyruvate that can be converted to acetyl-CoA, providing feedstock for fatty acid (FA) biosynthesis in plastid (*Niu et al., 2015a*). In plant, glycolysis occurs simultaneously in cytoplasm and plastid (*Niu et al., 2015b*). Previous study in oil palm has shown that plastidial glycolysis plays a major role in oil accumulation (*Bourgis et al., 2011*). Interestingly, transcriptome analysis in *Ricinus communis*, *Euonymus alatus* and *Tropaeolum majus* suggested a greater proportion of carbon flux through cytosolic glycolysis for FA biosynthesis (*Troncoso-Ponce et al., 2011*). To date, important questions in developing oilseeds remain regarding how the two glycolytic pathways allocate the carbon flux into FA biosynthesis.

To store oil in plant seeds, carbon source (acetyl-CoA) have to go through FA biosynthesis in plastid and triacylglycerol (TAG) assembly in endoplasmic reticulum (*Bates, Stymne & Ohlrogge, 2013*). Before entering the plastidial FA biosynthesis, acetyl-CoA musted be carbonylated to generate malonyl-CoA by acetyl-CoA carboxylase (ACC), and then acyl carrier protein (ACP) is transferred to replace CoA by malonyl-CoA: ACP malonyltransferase (MCMT) (*Baud & Lepiniec, 2009*). The extension of two-carbons in each cycle of FA biosynthesis until saturated 16:0-ACP is produced. Subsequently, the C16:0 chain can be extended to C18:0 by ketoacyl-ACP Synthase II (KASII), and then desaturated to C18:1 by stearoyl-ACP desaturase (SAD) (*Baud & Lepiniec, 2009*). These generated FA chains are transported from plastid to ER. Remarkably, C18:1 chain can be esterified to phosphatidylcholine (PC) in ER, and are desaturated to generate C18:2 and C18:3 by fatty acid desaturase 2 (FAD2) and FAD3, respectively (*Bates, Stymne & Ohlrogge, 2013*). In general, FAs can be assembled into TAG by classic Kennedy pathway with series of enzymes, including glycerol-3-phosphate acyltransferase (GPAT), lysophosphatidic acid acyltransferase (LPAAT), phosphatidate phosphatase (PP), and

diacylglycerol acyltransferase (DGAT) (*Bai et al., 2020*). Additionally, the FA chains in PC pool can be interchanged with the FA chains in diacylglycerol (DAG) by PC: DAG cholinephosphotransferase (PDCT), or can also be used for TAG generation by phospholipid: DAG acyltransferase (PDAT) (*Horn & Chapman, 2014*).

In recent years, next-generation sequencing (NGS) has become a powerful tool for accurate analysis of RNA transcripts (*Gong et al., 2020*; *Wang et al., 2012*). Previous NGS transcriptome studies in *P. volubilis* revealed that some candidate genes were involved in oil accumulation (*Hu et al., 2018*; *Wang & Liu, 2014*; *Wang et al., 2012*). However, due to the limitations of NGS technology, for example the sequencing sequence is too short, some candidate genes were inaccurately assembled, leading to unknown errors in the date analysis. With the development of the sequencing technology, single molecular real-time (SMRT) sequencing can overcome this limitation and eliminate assembled errors. Using PacBio platform, the length of average reads from SMRT sequencing is about 10 KB, and the sub-read length can reach 35 KB (*Roberts, Carneiro & Schatz, 2013*). However, the sequencing error rate of SMRT is high, which requires repeated sequencing to correct. Combination of NGS and SMRT sequencing has been widely used in plants to collect large-scale long-read transcripts (*Gong et al., 2020*; *Zhang et al., 2021*; *Zhu et al., 2020*).

Based on our previous investigation of *P. volubilis* germplasm resources, the superior plant-type (SPT) with high oil content and inferior plant-type (IPT) with low oil content were used as plant materials. Here, the differences in oil content and FA composition were compared between SPT and IPT at 10, 70, and 110 days after pollination (DAP). Through the combination of Illumina sequence and PacBio SMRT technologies, the bioinformatics data of developing PVS were analyzed, including functional annotation, gene ontology (GO), long non-coding RNA (lncRNA) prediction, and transcription factors (TF) identification. Moreover, we comparatively analyzed the expression profiles of genes associated with glycolysis, FA desaturation, and TAG assembly in developing PVS. The results provide a better understanding of oil accumulation in developing PVS, and may present strategies for engineering oil accumulation in other oilseeds.

## MATERIALS & METHODS

### Plant material

Based on the previous collection of *P. volubilis* germplasm resources (unpublished data), these *P. volubilis* plants were cultivated in the plantation base of Hainan University, Danzhou, Hainan, China (latitude and longitude: 109.503179, 19.542727). Also, two germplasms with significant differences in oil content were used plant material in this study. According to previous study on the fruit development of *P. volubilis* (*Niu et al., 2014*), the PVS were collected at 10, 70, and 110 DAP, representing early, middle, and late phases of oil accumulation, respectively. *P.volubilis* is characterized by continuous flowering and fruiting, and thus these fruits were simultaneously harvested.

### Oil extraction

PVS were dried to constant weight at room temperature. The 2.00 g PVS was crushed and extracted by n-hexane extraction for 8 hours at 45 °C using Soxhlet extraction system. The

oil content of PVS was determined by measuring the weight reduced after PVS extraction. The step was performed in triplicate.

## FA methyl ester analysis

The extracted oils were mixed with 5wt % heptadecanoic acid. Mixed samples were trans-esterified under standard conditions (Methanol: Oil = 5.5:1) by 1 wt% potassium hydroxide 65 °C for 1 h. After standing for 1 h, the mixture was separated into two layers. The supernatant was extracted and dried to obtain FA methyl esters (FAMEs). The FAMEs were analyzed by gas chromatography-mass spectrometer (GC-MS) using the Agilent 6890 equipped with a flame ionization detector (GB/T17377-1998). The following chromatographic conditions: inlet temperature 260 °C, detector temperature 280 °C, injection volume 1 μL, high-purity hydrogen carrier gas. The qualitative and quantitative analysis of FAMEs was identified by comparing their retention times and peak area with ANPEL FAMEs mix (ZZSRM, Shanghai, China) and heptadecanoic methyl ester (ANPEL, Shanghai, China), respectively. All results were repeated for three times.

## Library construction and SMRT sequencing

The total RNA was isolated by grinding PVS on dry ice with Trizol reagent Kit (Thermo Fisher, China, Shanghai). The integrity and concentrations of the RNA were determined using Agilent 2100 biological analyzer and Nanodrop micro-spectrophotometer (Thermo Fisher, Shanghai, China), respectively. The full-length cDNA of mRNA is synthesized by SMARTer$^{TM}$ PCR cDNA Synthesis Kit (Takara, Shiga, Japan). Then, the full-length cDNA was extended into double-stranded cDNA by large-scale PCR reactions. The cDNA was end repaired and then attached to a sequencing adapter. The SMRT bell template was linked to polymerase, followed using SMRT sequencing on the PacBio Sequel system (Guangzhou, China). Ultra-long read (median 10KB) can be obtained, which contain a single complete transcript sequence information, and no assembly is required for post-analysis.

The sequencing raw data were processed by SMRTlink 5.0 software. The analysis process of obtaining full-length transcriptome mainly includes 3 stages. The raw reads were processed into Circular consensus sequences (CCS) according to the adaptor. Next, full-length and non-chemiric (FLNC) transcripts were determined by searching for the poly(A) tail signal and the 5′ and 3′ cDNA primers in CCS. Finally, the full-length sequences of the same transcript are clustered, and the similar full-length sequences are clustered into a cluster, and each cluster can obtain a consistent sequence. Consistent sequences are corrected to obtain high-quality sequences for subsequent analysis. Sequencing was repeated three times.

## Illumina RNA-seq library construction and sequencing

Total RNA was extracted from each biological sample by Spectrum Plant Total RNA Kit (Sigma-Aldrich, USA) on dry ice. RNA concentration was determined with the Nanodrop micro-spectrophotometer (Thermo Fisher, Shanghai, China). RNA integrity was measured using the Bioanalyzer 2100 system (Agilent Technologies, CA, USA). Second, mRNA was enriched with magnetic beads with Oligo (dT), followed by Fragmentation Buffer was added to mRNA for random Fragmentation. Third, the first cDNA strand was synthesized

by random hexamers using mRNA as template, and dNTPs, RNase H and DNA Polymerase are added to synthesize second cDNA strand. The repaired and purified double-stranded cDNA was then added Poly-A and attached to the sequencing connector. Then, the size of the resulting fragments was selected using AMPure XP beads. Finally, the cDNA library was enriched by PCR amplification and sequenced on Illumina's HiSeq™4000 platform (Illumina, San Diego, CA, USA).

Clean reads were obtained by removing reads containing adapter, reads containing ploy-N and low-quality reads from raw reads of Illumina sequencing. Although the reads of SMRT sequencing are longer, the sequence accuracy was lower than that of Illumina sequencing. To obtain the most accurate reference sequence, the clean reads of Illumina sequencing was used to correct the results of SMRT sequencing. These clean reads were aligned to the Pacbio reference sequences using Hisat2 tool (daehwankimlab.github.io/hisat2/). According to the comparison statistics, the base at this position of Pacbio sequence was replaced with the base with the most counts.

### Functional annotation of transcripts

Using BLAST software (version 2.2.26), the gene sequences were compared with the following databases: non-redundant protein sequence database (NR; http://www.ncbi.nlm.nih.gov), SwissProt (http://www.expasy.org/sprot), kyoto encyclopedia of genes and genomes (KEGG; http://www.genome.jp/kegg), gene ontology (GO; http://www.geneontology.org), cluster of orthologous groups of proteins (COG; http://www.ncbi.nlm.nih.gov/COG), and PFAM (https://pfam.xfam.org/), to obtain annotated information. ITAK software was used to predict transcription factors (TFs) and assign genes to different families (*Zheng et al., 2016*).

### Prediction and analysis of lncRNAs

Four computational approaches include CPC/CNCI/CPAT/Pfam/ were combined to sort non-protein coding RNA candidates from putative protein-coding RNAs in the transcripts. By a minimum length and exon number threshold, putative protein-coding RNAs were filtered out. Then, transcripts with the ability to distinguish protein-coding genes from non-coding genes were further screened using CPC/CNCI/CPAT/ PFAM. Based on base pairing between lncRNA and mRNA sequences, the potentially lncRNA-targeted transcripts were predicted using LncTar tool (*Li et al., 2015*).

### Differential expression

Gene expression levels were estimated by TPM (Transcripts Per Kilobase Million) using RSEM software (*Li & Dewey, 2011*). Differential expression analysis of two conditions/groups was performed using TBtools (*Chen et al., 2020*). Benjamini and Hochberg's method was used to adjust the result *P* value to control the error detection rate. The FDR <0.01 and foldchange ≥ 2 transcript was identified as differential expression genes (DEGs).

### Gene expression analysis

Based on sequencing results, the key gene expressions in different tissues were verified by RT-qPCR. According to previous investigation, *ubiquitin-conjugating enzyme* (*UCE*)

was used as the stable reference gene in different *P. volubilis* tissues (*Niu et al., 2015c*). The RT-qPCR primers were designed by PrimerQuest™ Tool (Table S1). The RT-qPCR reaction system was prepared using the MonAmp™ ChemoHS qPCR Mix Kit (Monad, Guangzhou, China). The RT-qPCR procedure were performed by Lightcycler 96 (Roche, Penzberg, Germany). The expression levels were calculated by $2^{-\Delta\Delta Ct}$.

## RESULTS AND DISCUSSION

### Oil content in developing PVS

Based on previous investigation on the fruit development and oil accumulation of *P. volubilis* (*Niu et al., 2014*), PVS were collected at 10, 70, and 110 DAP, which represented early, middle, and late stage of oil accumulation, respectively (Fig. 1A). The size of SPT and IPT seeds showed a similarly dynamic pattern, rapid rise from 10 to 70 DAP and slight decline 70 to 110 DAP (Fig. 1B). This slight decline of size may be due to dehydration behavior of seeds. As expected, the dry weight of SPT and IPT seeds exhibited a gradual increase (Fig. 1C), indicating the accumulation of stored substances with maturity. At mature stage (110 DAP), the PVS episperm disappeared and only brown endopleura was observed (Fig. 1A). There was no significant difference in seed size or dry weight between SPT and IPT during PVS development (Figs. 1B and 1C). These developmental characteristics are consistent with previous investigations in *P. volubilis* fruits (*Niu et al., 2014*; *Wang & Liu, 2014*).

In general, oil content and FA composition are the vital metric for extension and utilization of oilseeds. The oil contents of SPT (2.69% to 52.60%) and IPT (2.42% to 39.13%) both exhibited a gradual accumulation with PVS development (Fig. 1D), as was reported in *P. volubilis* fruits (*Niu et al., 2014*; *Wang & Liu, 2014*). The oil content of SPT seeds was significantly higher than that of IPT seeds at 70 and 110 DAP (Fig. 1D). Clearly, SPT will be more commercially viable. The significant difference conducive to further investigation on the regularly mechanism of oil accumulation in developing PVS.

### FA composition in developing PVS

Seven FAs were detected in developing PVS, including palmitic acid (C16:0), palmitoleic acid (C16:1), stearic acid (C18:0), oleic acid (C18:1), linoleic acid (C18:2), linolenic acid (C18:3), and eicosanoic acid (C20:0). In early development of seeds, the most abundant component was palmitic acid (C16:0), followed by stearic acid (C18:0), linoleic acid (C18:2), and linolenic acid (C18:3) (Table 1). As reported previously (*Niu et al., 2014*; *Wang & Liu, 2014*), linoleic acid (C18:2) linolenic acid (C18:3) were the predominant FAs in mature seeds (Table 1). Although all seven of FAs showed a pattern of gradual accumulation during PVS development, the cumulative rates of linoleic acid (C18:2) linolenic acid (C18:3) were the fastest (Table 1). It was worth noticing in mature seeds (110 DAP) that the C18:3 content in SPT was obviously higher (1.11 times as high) than in IPT (Table 1). This resulted in the oil content of SPT was significantly higher than that of IPT in mature seeds (Fig. 1D). Obviously, the more C18:3 in seed oil of SPT would provide even more value to the edible oil. This notable difference between SPT and IPT is more beneficial for us to further study the biosynthesis of linolenic acid.

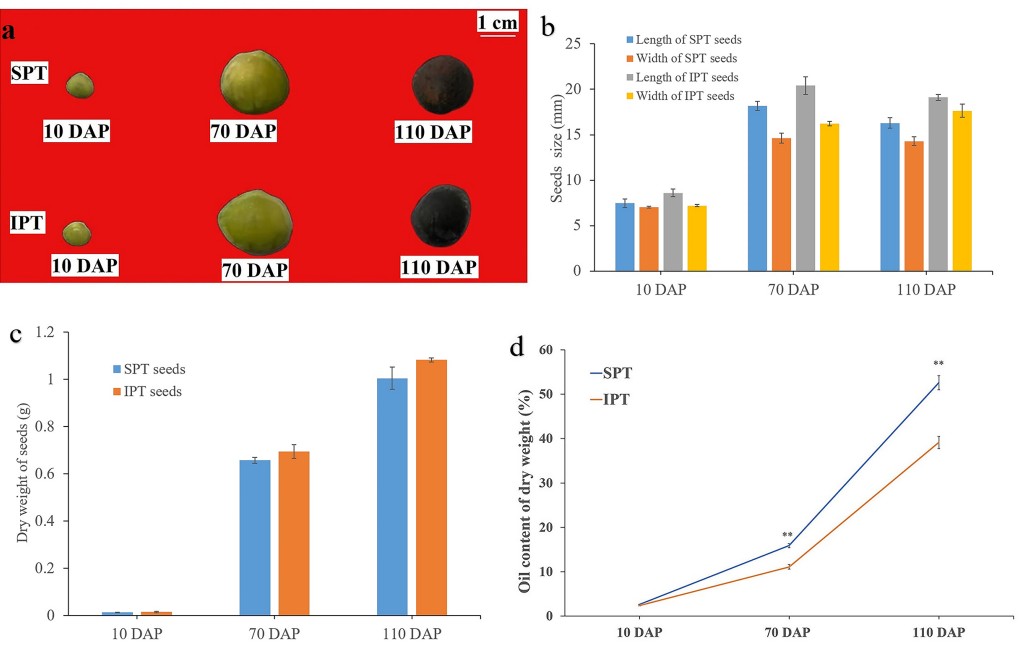

**Figure 1** **Developmental changes in size, weight, and oil content of PVS.** (A) Intact SPT and IPT seeds from 10, 70, and 110 DAP. (B) Changes in fresh weight of the developing PVS. (C) Changes in size of the developing PVS. (D) Comparative analysis of oil content between SPT and IPT seeds. Values are mean ± standard deviation. Asterisks (**) indicate statistical significance oil contents (30DAF, $P = 0.0137$; 70 DAF, $P = 0.0006$; 110 DAF, $P = 0.0008$).

**Table 1** **Dynamic changes of FA compositions in developing PVS.**

| | | Fatty acid composition (mg/g) | | | | | | |
|---|---|---|---|---|---|---|---|---|
| | | Palmitic acid (C16:0) | Palmitoleic acid (C16:1) | Stearic acid (C18:0) | Oleic acid (C18:1) | Linoleic acid (C18:2) | Linolenic acid (C18:3) | Eicosanoic acid (C20:0) |
| SPT | 10 DAP | 10.23 ± 1.20 | 0 ± 0.00 | 6.31 ± 0.71 | 1.35 ± 0.12 | 4.11 ± 0.51 | 4.01 ± 0.26 | 0 ± 0.00 |
| | 70 DAP | 12.12 ± 1.41 | 0 ± 0.00 | 9.45 ± 0.89 | 10.45 ± 1.23 | 34.89 ± 1.48 | 69.01 ± 2.64 | 0 ± 0.00 |
| | 110 DAP | 26.56 ± 2.78 | 1.31 ± 0.21 | 18.85 ± 1.31 | 52.39 ± 3.21 | 176.54 ± 6.43 | 223.45 ± 3.84 | 1.57 ± 0.12 |
| IPT | 10 DAP | 9.89 ± 1.15 | 0 ± 0.00 | 5.78 ± 0.41 | 1.26 ± 0.11 | 3.89 ± 0.26 | 3.74 ± 0.37 | 0 ± 0.00 |
| | 70 DAP | 11.21 ± 1.27 | 0 ± 0.00 | 8.06 ± 0.94 | 9.82 ± 1.23 | 32.01 ± 1.79 | 33.63 ± 2.18 | 0 ± 0.00 |
| | 110 DAP | 26.54 ± 2.03 | 1.22 ± 0.15 | 16.12 ± 1.35 | 55.84 ± 2.64 | 167.02 ± 5.10 | 105.76 ± 2.15 | 0.82 ± 0.10 |

## Sequencing and data processing

Integrated PacBio SMRT and Illumina technologies were employed for transcriptomic analysis in developing PVS. The raw data were stored in the NCBI SRA database (access number: PRJNA744032). The 50 Gb of Illumina data were gained from developing PVS, and each set of data had a sound quality with Q30 ≥ 95% (Table S2). Additionally, a total of 109 Gb clean data was obtained from PacBio SMRT sequencing. Notably, it was the first full-length transcriptomic sequencing in *P. volubilis*. The screening of subreads produced 461,967 CCS (746,599,639 read bases) with an average read length of 1,616 bp (Fig. S1a). By testing whether the CCS sequences contain a 5′ primer, 3′ primer, and the poly-A

tail, 398,863 FLNC (86.34%) were identified with a mean length of 1,179 bp (Fig. S1b). Similar sequences from FLNC were clustered together by SMRTLink software, resulting in 157,193 uncorrected consensus isoforms with a mean length of 1,482 bp (Fig. S1c). After correcting consensus isoforms, 157,177 polished high-quality isoforms were screened out. In the process of full-length transcription cluster, redundant sequences are inevitably generated. After removing redundancy by CD-HIT (*Li & Godzik, 2006*), a total of 68,971 corrected isoforms were obtained with a mean read length of 1926 bp and an N50 of 2547 bp. By a significant reduction in the number of non-redundant full-length transcripts, the depth and integrity of the transcriptome was significantly improved using SMRT long-read sequencing.

Previous studies on *P. volubilis* transcriptome were only based on NGS (*Hu et al., 2018*; *Wang et al., 2012*). For example, 70,392 unigenes (average length of 645 bp) were assembled from two developmental stages of PVS (*Wang et al., 2012*). Also, 124,750 non-redundant transcripts (average length of 851 bp) were obtained from eight organs of *P. volubilis* (*Hu et al., 2018*). However, the limitation of short reading and assemble error may lead to inaccuracy and difficulty to bioinformatics analysis (*Au et al., 2012*). SMRT sequencing can capture reads with mean length of over 2500-10,000 bp, and combine high-accuracy Illumina reads to obtain accurate transcriptome (*Korlach et al., 2017*). Combination of SMRT and Illumina sequencing, we obtained transcripts with a mean read length of 1926 bp, which is significantly greater than the 645 bp (*Wang et al., 2012*) and 851 bp (*Hu et al., 2018*) in previous studies. This implied that a more precise and complete full-length transcriptome was acquired, contributing to the expansion of further analysis.

## Functional annotation of the full-length transcriptome of *P. volubilis*

According to the results of functional annotation, 61,082 (88.56%), 43,501 (63.07%), 23,913 (34.67%), 29,287 (42.46%) genes were annotated in the NR, SwissProt, PFAM, and KEGG databases, respectively (Fig. S2a and Table S3). Among the NR database, most of genes (38,421, 62.92%) were homologous to *R. communis* that belongs to the same family Euphorbiaceae with *P. volubilis* (Fig. S2b), indicating the reliability of our transcriptome sequencing. Based on the GO annotation, 47,609 genes (77.78%) were annotated (Table S3). The "binding" (23,773), "cell" (22,698), and "metabolic process" (24,495) were the most abundant GO terms in molecular function, cellular component, and biological process, respectively (Fig. S3a). As for COG annotation, 39,912 genes (55.43%) were divided into 26 subclasses, of which "general function prediction only" accounted for the highest percentage (Fig. S3b). KEGG annotation identified 29,287 genes in 126 pathways (Table S3). Importantly, several pathways involved in carbon source, FA biosynthesis, elongation and degradation, and TAG assembly were identified, such as 573 genes in "glycolysis/gluconeogenesis (ko00010)", 218 genes in "fatty acid biosynthesis (ko00061)", 81 genes in "fatty acid elongation (ko00062)", 225 genes in "fatty acid degradation (ko00071)", and 243 genes in "glycerolipid metabolism (ko00561)" (Table S3). Taken together, the high-quality results of transcriptome annotation can help to further explore the mechanism of oil accumulation in developing PVS.

## Analysis of differential gene expression in developing PVS

The clean reads were mapped to non-redundant transcripts to calculate the digital expressions. If using reads or fragments per kilobase per million, the sum of the normalized reads for each sample may be different, making it difficult to compare samples directly (*Zhao, Ye & Stanton, 2020*). However, when calculating TPM, first standardizes the length of genes and then the sequencing depth. The sum of all TPMs in each sample is the same, which is a better measure to compare samples. Thus, TPM algorithm was applied to calculate the expression levels of genes to acquire accurate DEGs (Table S4). In addition, the Pearson correlation coefficient was calculated to evaluate the correlation of biological repeats. An effective approach can test the repeatability of biological repeats, evaluate the reliability of DEGs, and assist in screening abnormal samples. The high intergroup variability and intragroup correlation indicated notable variation and good repeatability in intergroup and intragroup, respectively (Fig. 2A).

Using DESeq2 software, a total of 11,178 DEGs were obtained from pairwise comparison, among which IPT-10 *vs* IPT-110 had the most DEGs (2,136 up-regulation and 3,332 down-regulation), and IPT-110 *vs* SPT-110 possessed the least DEGs (274 up-regulation and 305 down-regulation) (Fig. 2B). Subsequently, we performed GO analysis to explore the functional cluster of these DEGs. In biological process, it is interesting that DEGs is mainly concentrated in the "carbohydrate metabolic process" (GO:0005975), "protein metabolic process" (GO:0019538), and "regulation of primary metabolic process" (GO:0080090) cluster by pair comparison (Fig. S4). The resulting implied an intricate regulatory mechanism of carbon flux for oil biosynthesis in developing PVS.

## LncRNA identification and function prediction

It has been reported that lncRNAs play crucial regulatory roles in numerous biological processes (*Wu et al., 2020*). To predict lncRNAs in developing PVS, the protein-coding capacities of transcripts were judged according to public protein databases. After filtering out the genes with potential coding sequences, 7,823 high-confidence lncRNAs were shared among four databases (Table S5). LncRNAs have been shown to be involved in diverse aspects of post-transcriptional gene regulations in plants, especially in response to the environment, which may be a key role in plant adaptation during evolution (*Fonouni-Farde, Ariel & Crespi, 2021*). Based on base pairing between lncRNA and mRNA sequences, the 7,798 lncRNA-targeted transcripts were predicted using the LncTar tool (*Li et al., 2015*) (Table S5). Interestingly, 594, 811, 221, 733, and 172 lncRNAs were implicated in "glycolysis/gluconeogenesis", "fatty acid biosynthesis", "fatty acid elongation", "fatty acid degradation", and "glycerolipid metabolism", respectively (Fig. S5). This result suggests that the predicted lncRNAs may participate in the potential regulation of oil accumulation during PVS development.

## TF identification and correlation analysis

In plant growth and development, TF is a vital component of transcriptional regulation system (*Meraj et al., 2020*). To comprehensively explore TFs in developing PVS, the ITAK software was used to identify TFs (*Zheng et al., 2016*). A total of 5,830 TFs from 209 TF

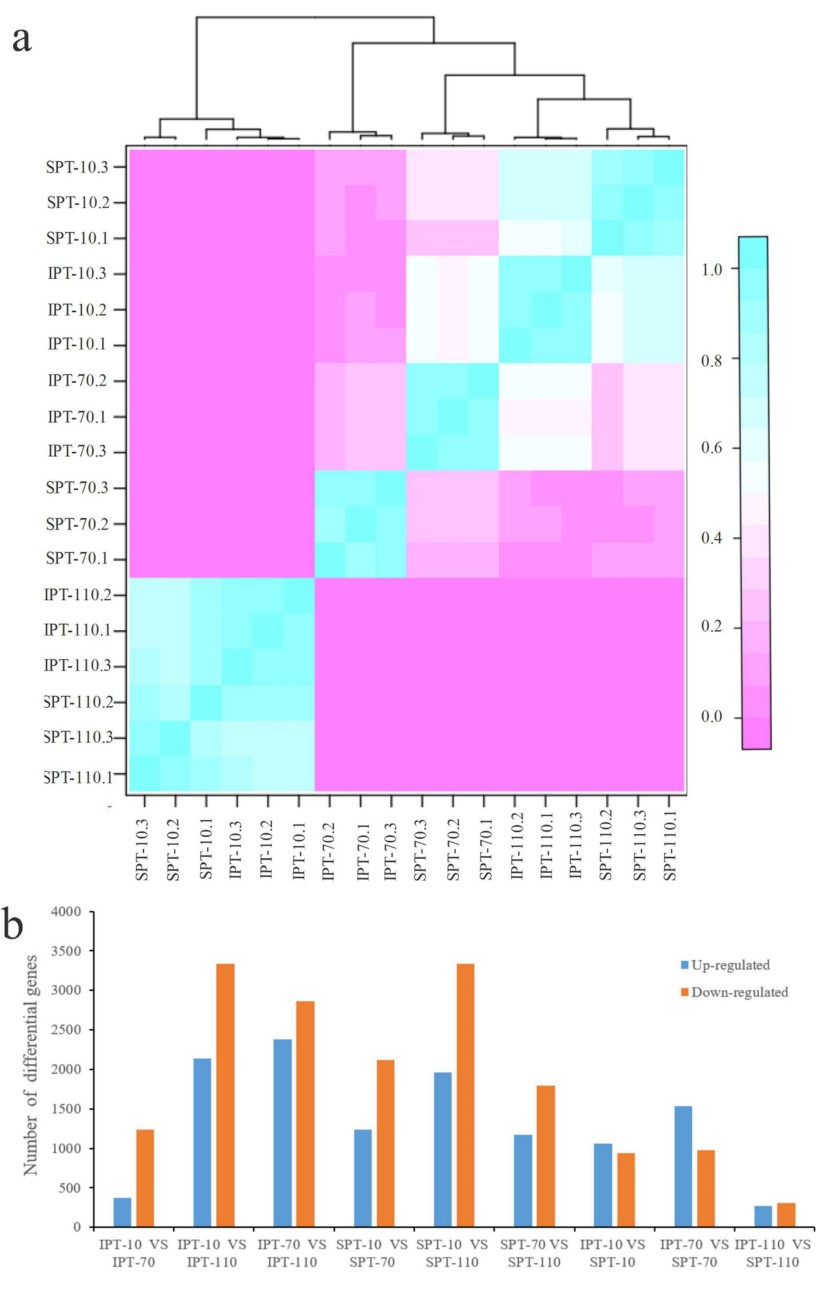

**Figure 2** **Analysis of differential gene expression in developing PVS.** (A) Pearson correlation coefficient of transcriptome samples. The greater the value, the greater the correlation. (B) The distribution of DEGs in different groups.

families were identified, mainly including C3H (167), bHLH (159), MYB-related (159), bZIP (128) (Table S6). Among these TFs, TFs with low expression levels (TPM<5) were excluded from the following analysis. To explore which TFs are potentially related to oil accumulation in developing PVS, the correlation analysis of expression patterns and oil

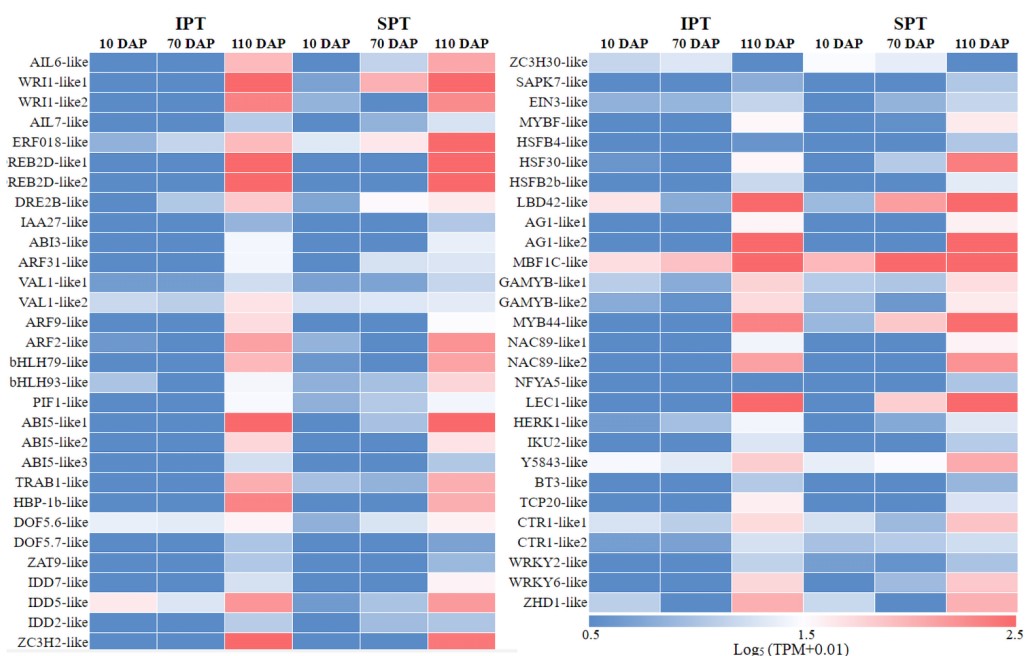

**Figure 3** **The expression heatmap of key TFs positively correlated with oil content in developing PVS.**

contents was performed (Table S6). The resulting 58 TFs was identified to be positive correlation with oil accumulation in developing PVS (Fig. 3).

Currently, the widely crucial TFs in oil biosynthesis were WRINKLED1 (WRI1), FUSCA3 (FUS3), ABSCISIC ACID INSENSITIVE3 (ABI3), LEAFY COTYLEDON1 (LEC1) and LEC2 (*Bates, Stymne & Ohlrogge, 2013*; *Baud & Lepiniec, 2009*). As an APETALA2 (AP2)-type TF, WRI1 plays a central role in the overall metabolic regulatory network of oil accumulation, activating genes related to glycolysis and FA biosynthesis (*Deng et al., 2019*). Two WRI1 homologues were identified in developing PVS, and their expression levels exhibited significant positive correlation with oil accumulation (Fig. 3 and Table S6). This suggests the importance of WRI1 in oil accumulation during PVS development. The expression levels were nearly twice as much as in *WRI1-like1* when compared to *WRI1-like2* in 110 DAP (Table S6). Notably, the *WRI1-like1* expression for SPT in 70 DAP was notably higher than that for IPT (Fig. 3), which was was confirmed by RT-qPCR data (Fig. 4). It has been confirmed that WRI1 participates in directing the carbon flux toward FA biosynthesis (*Bates, Stymne & Ohlrogge, 2013*; *Deng et al., 2019*). Combined with the results of oil content in developing PVS (Fig. 1D), earlier transcription of WRI1 may play a circular role in directing the carbon flux that enters PVS oil accumulation.

LEC1, a family of NF-YB transcription factors, as an upstream regulator of WRI1, is involved in seed development and oil accumulation (*Jia, Suzuki & McCarty, 2014*). Indeed, our study found that the expression levels of *LEC1-like* gene showed a similar trend with *WRI1-like1* gene (Fig. 3), as was shown in the RT-qPCR results (Fig. 4). These expression data suggested that LEC1-like may be an upstream regulator of WRI1-like1 in developing

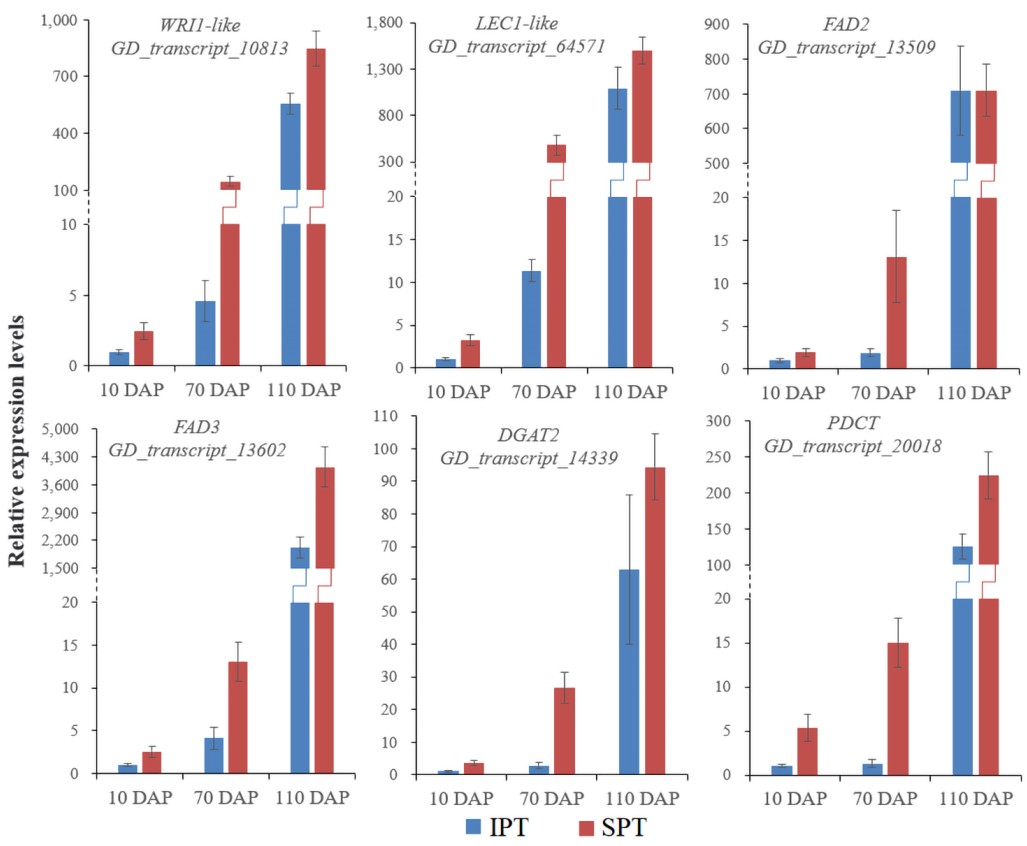

**Figure 4  RT-qPCR analysis of some key genes involved in oil accumulation.** The expression levels were calculated by $2^{-\Delta\Delta Ct}$.

PVS. Although the expressions of LEC1-like was not substantial difference between SPT and IPT in 110 DAP, the expression of LEC1-like1 in SPT was 162-fold higher than that of IPT in 70 DAP (Table S6). As discussed above, once PVS complete morphogenesis, the sooner transcriptional regulation of TF (related to oil accumulation) begins, the more conducive to oil accumulation.

Despite the homologues of LEC2 and FUS3 have been identified in developing PVS, the TPM levels of *LEC2* and *FUS3* genes were not positively correlated with oil content (Table S6). Thus, it is speculated that they may be not the core regulators of oil accumulation in developing PVS. In the MYB family, the MYB96 transcription factor has been reported to promote oil accumulation (*Seo et al., 2009*). It was found that the expressions of *MYB44-like* showed positive correlation with oil accumulation in developing PVS (Table S6). Moreover, the expression of *MYB44-like* in SPT was 40-fold higher than that of IPT in 70 DAP (Fig. 3). Thus, MYB44-like may actively participate in the regulation of oil accumulation in developing PVS.

## Pyruvate generation for FA biosynthesis

In plants, glycolysis yields pyruvate necessary to synthesize FA, and occurs both in cytosol and plastid (*Plaxton, 1996*). Indeed, the homologous enzymes catalyzing the nine steps of glycolysis were found in both cytosol and plastid (Table S7). Previously, the investigation on expression profiles of oil palm implied that plastidial glycolysis plays a major role in oil accumulation (*Bourgis et al., 2011*). However, the expression analysis in *R. communis*, *E. alatus* and *T. majus* suggested a greater proportion of carbon flux through cytosolic glycolysis for FA biosynthesis (*Troncoso-Ponce et al., 2011*). Hexokinase (HK), 6-phosphofructokinase (PFK) and pyruvate kinase (PK) are considered as to rate-limiting enzymes in glycolysis (*Plaxton, 1996*). In developing PVS, their TPM levels of plastidial isoforms were significantly lower than that of cytosolic isoforms in middle (70 DAP) or late (110 DAP) stages of oil accumulation (Fig. 5). Also, some cytosolic genes, encoding fructose-bisphosphate aldolase (FBA), triosephosphate isomerase (TIM), glyceraldehyde 3-phosphate dehydrogenase (GAPC), and phosphoglycerate kinase (PGK), exhibited higher expression levels than plastidial isoforms (Fig. 5). These expression results in developing PVS indicated more active transcription of genes in cytosolic glycolysis, implying a strong flux of carbon toward pyruvate through cytosolic glycolysis.

## Characterization of key genes involved in FA desaturation and TAG assembly

The storage oils of IPT and SPT were characterized by a high (>75%) proportion of polyunsaturated FAs (PUFAs), but the most abundant PUFA was different (Table 1). For PUFA biosynthesis in PC pool of endoplasmic reticulum, oleic acid (C18:1) can be desaturated to form linoleic acid (C18:2) by omega-6 FA desaturase 2 (FAD2) and further desaturated to generate linolenic acid (C18:3) by FAD3 (*Mai et al., 2020*). A total of five *FAD2* and nine *FAD3* genes were identified in this study (Fig. 6). Owing to the integrity of third-generation transcriptome sequencing, these identified gene numbers were larger than previous transcriptome reporters (*Hu et al., 2018*; *Liu et al., 2020*; *Wang et al., 2012*).

Intrudingly, different *FAD2* genes showed differently temporal expression profiles in developing PVS, as was reported in previous transcriptome analysis (*Liu et al., 2020*). The GD_transcript_109520 was drastically up-regulated at 10 and 70 DAP, whereas GD_transcript_13509 was significantly up-regulated only at 110 DAP (Figs. 4 and 6). Given that a high percentage of C18:2 in developing PVS (Table 1), these temporal expression patterns suggest that different *FAD2* genes may operate at different developmental stages for C18:2 accumulation. Although nine homologues of FAD3 were identified in developing PVS, only GD_transcript_13602 exhibited a notable up-expression at 110 DAP (Fig. 6). This temporal expression pattern in developing PVS was comparable to previous report (*Liu et al., 2020*). Form our RT-qPCR results, this *FAD3* expression was higher in SPT than in IPT at 70 and 110 DAP (Fig. 4), matched by high and low C18:3 content, respectively (Table 1).

As for the genes involved in TAG assembly, the orthologs of PDCT (GD_transcript_20018), PDAT (GD_transcript_125033), and DGAT (GD_transcript_14339) in particular showed up-regulated expression at 110 DAP (Fig. 6), similar to the dynamic trend of oil

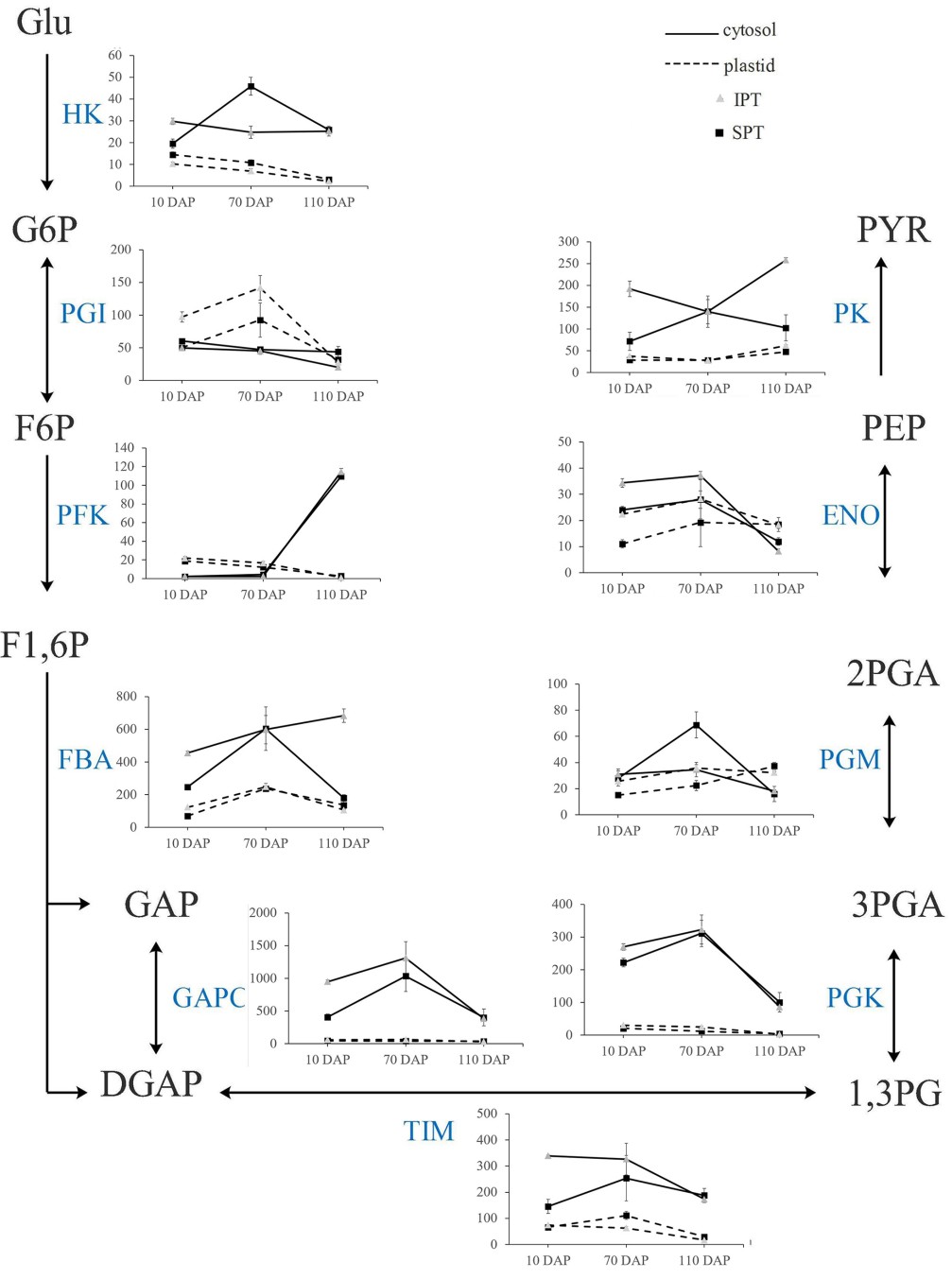

**Figure 5 Transcript patterns for key enzymes involved in glycolysis.** In the line chart, the ordinate represents the TPM levels. Abbreviation: HK, hexokinase; PGI, Phosphoglucose isomerase; PFK, 6-phosphofructokinase; FBA, fructose-bisphosphate aldolase; GAPC, glyceraldehyde 3-phosphate dehydrogenase; TIM, triosephosphate isomerase; (continued on next page...)

**Figure 5 (...continued)**
PGK, phosphoglycerate kinase; PGM, 2,3-bisphosphoglycerate-dependent phosphoglycerate mutase; ENO, enolase; PK, Pyruvate kinase; Glu, $\alpha$-D-Glucose; G6P, $\alpha$-D-Glucose-6-phosphate; F6P, $\beta$-D-Fructose-6-phosphate; F1,6P, $\beta$-D-Fructose-1,6-diphosphate; GAP, Glycerone-phosphate; DGAP, Glyceraldehyde-3-phosphate; 1,3PG, Glycerate-1,3-diphosphate; 3PGA, Glycerate-3 -phosphate; 2PGA, Glycerate-2-phosphate; PEP, Phosphoenol-pyruvate; PYR, Pyruvate.

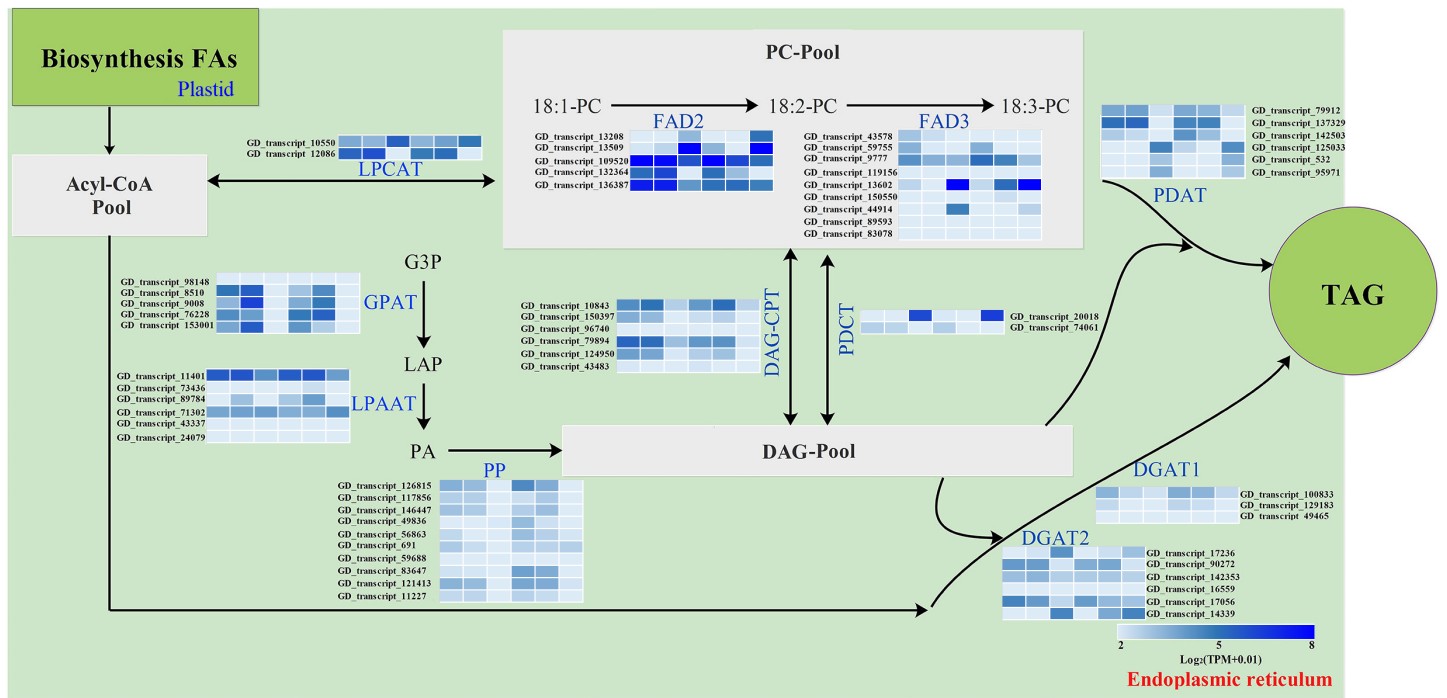

**Figure 6** **Identification of genes related to FA desaturation and TAG assembly in developing PVS.** The heatmap from left to right represent the data from IPT-10 DAP, IPT-70 DAP, IPT-110 DAP, SPT-10 DAP, SPT-70 DAP, SPT-110 DAP, respectively. Abbreviations: LPCAT, Lysophospholipid acyltransferase; FAD, Fatty acid desaturase; GPAT, Glycerol-3-phosphate acyltransferase; LPAAT, 1-acylglycerol-3-phosphate O-acyltransferase; PP, Phosphatidate phosphatase; DAG-CPT, Diacylglycerol cholinephosphotransferase; PDCT, Phosphatidylcholine:diacylglycerol cholinephosphotransferase; PDAT, Phospholipid:diacylglycerol acyltransferase; DGAT, Diacylglycerol O-acyltransferase; PC, phosphatidylcholine; G3P, glycerol-3-phosphate; LPA, 1-acylglycerol-3-phosphate; PA, 1,2-diacylglycerol-3-phosphate; DAG, 1,2-diacylglycerol; TAG, triacylglycerol.

accumulation (Fig. 1). Therefore, it is reasonable to presume that these genes may play an important role in oil accumulation during PVS development. It was reported that two unrelated types of DGAT enzymes have been confirmed to associated with TAG generation (*Lardizabal et al., 2001*). DGAT1 is the predominant enzyme synthesizing TAG in Arabidopsis seeds (*Zou et al., 1999*), whereas the expression levels of DGAT2 far exceed that of DGAT1 observed in *R. communis* (*Troncoso-Ponce et al., 2011*). In this work, the ortholog of DGAT2 (GD_transcript_14339) showed higher levels of expression than the orthologs of DGAT1 (Fig. 6). Also, GD_transcript_14339 showed up-regulated expression at 110 DAP by TPM and RT-qPCR analysis (Figs. 4 and 6). Therefore, DGAT2 may be responsible for TAG synthesis in developing PVS. In plants, PDCT can exchange acyl chains between PC and DAG (*Lu et al., 2009*). Previous expression profiles in *Prunus sibirica* (*Niu*

*et al., 2015a*) and *Lindera glauca* (*Niu et al., 2015b*) indicated a down-regulated expression of *PDCT* gene during seed development. The storage oils of *P. sibirica* and *L. glauca* were characterized by a high (>60%) proportion of saturated or monounsaturated FAs (*Niu et al., 2015a*; *Niu et al., 2015b*). Intriguingly, the *PDCT* gene (GD_transcript_20018) exhibited notably up-regulated expression at mature PVS seed (110 DAP), and its expression levels showed significantly higher in SPT than in IPT (Figs. 4 and 6). Our results imply that PVS likely utilize PDCT activity to accumulate PUFA (especially C18:3) in TAG, providing leads for future research on the interconnection of pathways involved in PC and TAG synthesis.

## CONCLUSIONS

The investigations for oil content and FA composition in developing PVS indicated that the most abundant FAs in SPT and IPT seeds was polyunsaturated FA (linoleic acid and linolenic acid). Importantly, the main difference was that the C18:3 content of SPT was 1.11-fold higher than that of IPT, resulting in the higher oil content in mature SPT seeds. Full-length and next-generation transcriptome sequencing identified 68,971 non-redundant genes, of which 61,217 (88.76%) genes were annotated in at least one database. By differential expression analysis, 11,178 DEGs was identified to be mainly related to carbon mechanism in developing PVS. A total of 7,823 lncRNAs and 7,798 lncRNA-targeted genes were predicted. In addition, 5,830 TFs was identified to belong to 209 TF families. Notably, the expression profiles of 58 TFs exhibited a significantly positive correlation with oil contents in developing PVS. Also, the analysis of expression profiles suggested that FAD3, PDCT, PDAT, and DAGT2 may play important roles in biosynthesis and assemble of linoleic acid. These findings improve our understanding of PUFA biosynthesis in developing PVS.

**Abbreviations**

| | |
|---|---|
| **PFK** | 6-phosphofructokinase |
| **ABI3** | ABSCISIC ACID INSENSITIVE3 |
| **ACC** | acetyl-CoA carboxylase |
| **ACP** | acyl carrier protein |
| **AP2** | APETALA2 |
| **CCS** | circular consensus sequences |
| **DAP** | days after pollination |
| **DAG** | diacylglycerol |
| **DGAT** | diacylglycerol acyltransferase |
| **DEG** | differential expression gene |
| **ER** | endoplasmic reticulum |
| **FAD** | FA desaturase |
| **FAME** | FA methyl ester |
| **FA** | fatty acid |
| **FBA** | fructose-bisphosphate aldolase |
| **FLNC** | full-length and non-chemiric |
| **FUS3** | FUSCA3 |

| | |
|---|---|
| **GC-MS** | gas chromatography-mass spectrometer |
| **GAPC** | glyceraldehyde 3-phosphate dehydrogenase |
| **GPAT** | glycerol-3-phosphate acyltransferase |
| **HK** | Hexokinase |
| **IPT** | inferior plant-type |
| **KASII** | ketoacyl-ACP Synthase II |
| **LEC1** | LEAFY COTYLEDON 1 |
| **lncRNA** | long non-coding RNA |
| **LPAAT** | lysophosphatidic acid acyltransferase |
| **MCMT** | malonyl-CoA: ACP malonyltransferase |
| **NGS** | next-generation sequencing |
| **PVS** | *P. volubilis* seed |
| **PDCT** | PC: DAG cholinephosphotransferase |
| **PP** | phosphatidate phosphatase |
| **PC** | phosphatidylcholine |
| **PGK** | phosphoglycerate kinase |
| **PDAT** | phospholipid: DAG acyltransferase |
| **PUFA** | polyunsaturated FA |
| **PK** | pyruvate kinase |
| **SMRT** | single molecular real-time |
| **SAD** | stearoyl-ACP desaturase |
| **SPT** | superior plant-type |
| **TF** | transcription factor |
| **TAG** | triacylglycerol |
| **TIM** | triosephosphate isomerase |
| **WRI1** | WRINKLED1 |

### Funding
This work was supported by the Hainan University Research Project (KYQD(ZR)-22056 and KYQD(ZR)20055). The funders had no role in study design, data collection and analysis, decision to publish, or preparation of the manuscript.

### Grant Disclosures
The following grant information was disclosed by the authors:
Hainan University Research Project: KYQD(ZR)-22056, KYQD(ZR)20055.

### Competing Interests
The authors declare there are no competing interests.

## Author Contributions

- Yijun Fu performed the experiments, analyzed the data, prepared figures and/or tables, authored or reviewed drafts of the article, and approved the final draft.
- Kaisen Huo performed the experiments, analyzed the data, prepared figures and/or tables, authored or reviewed drafts of the article, and approved the final draft.
- Xingjie Pei performed the experiments, authored or reviewed drafts of the article, and approved the final draft.
- Chongjun Liang performed the experiments, authored or reviewed drafts of the article, and approved the final draft.
- Xinya Meng conceived and designed the experiments, authored or reviewed drafts of the article, and approved the final draft.
- Xiqiang Song conceived and designed the experiments, authored or reviewed drafts of the article, and approved the final draft.
- Jia Wang conceived and designed the experiments, analyzed the data, prepared figures and/or tables, authored or reviewed drafts of the article, and approved the final draft.
- Jun Niu conceived and designed the experiments, analyzed the data, prepared figures and/or tables, authored or reviewed drafts of the article, and approved the final draft.

## DNA Deposition

The following information was supplied regarding the deposition of DNA sequences:

The sequence data is available at NCBI: PRJNA744032.

## Data Availability

The raw data is available in the Supplemental Files.

## Supplemental Information

Supplemental information for this article can be found online at http://dx.doi.org/10.7717/peerj.13998#supplemental-information.

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
