# Peer review of "Full-length transcriptome revealed the accumulation of polyunsaturated fatty acids in developing seeds of Plukenetia volubilis"

_PeerJ, doi:10.7717/peerj.13998_

## Round 0.1 · original submission · Major Revisions

Although the manuscript has some valuable information but it requires major revision.

Reviewer 1 ·

Basic reporting

The manuscript number #71870 reports “Full-length transcriptome revealed the accumulation of polyunsaturated fatty acids in developing seeds of Plukenetia volubilis”. Full-length transcriptome of two plant type of Plukenetia volubilis, superior plant-type (SPT) and inferior plant-type (IPT), were analyzed with based on the difference of the oil content of seeds. Some candidate genes involved in oil biosynthesis or the accumulation of polyunsaturated FAs of P. volubilis were screened. However, there are some questions need to be clarified and further improved in the MS.

Experimental design

Point 1: Oil is mainly stored in kernels, it will be clear and convincing that the dry weight of kernels is compared between SPT and IPT, instead of fresh weight of seeds in Fig. 1b. Development status and moisture content of kernels might affect the result of oil content.
Point 2: Authors should select some candidate genes related oil biosynthesis and the accumulation of polyunsaturated FAs to verify the transcriptome results with quantitative realtime PCR analysis, such as WRI1-like1, FAD2, FAD3, DGAT2, and etc.

Validity of the findings

Point 3: In Conclusions, “the main difference was that the C18:3 content of SPT… , resulting in the higher oil content in mature SPT seeds” in fact, the difference of the C18:2 content is similar to the C18:3 content between SPT and IPT (Table 1), so the C18:2 content should also be described. On the other hand, the difference of the expression levels of FAD2 and FAD3 were not significant between SPT and IPT in transcriptome results. It will be better that the difference of the expression levels of FAD2 and FAD3, respectively, can be detected at some stage of the seeds in SPT and IPT with quantitative realtime PCR analysis.
Point 4. Is the information of SPT and IPT missing in Table 1? I can only speculate which is SPT or IPT, please add it.

·

Basic reporting

In general the manuscript is well written, and the English used is tecnically correct.
The results presented are new, and contribute to the current knowledge about the accumulation of fatty acids of Plukenetia volibilis.
The literature referecnes are considered as apropriate.
The structure of the manuscript is standard and the number of tables and figures is in agreement with the results presented.

Experimental design

The way in which the experiments were carried out is correct and demonstrate scientific rigurosity.
The methods were all well described with enough information to be reproducible.
The data were statistically analyzed.

Validity of the findings

The results presented by the authors are new, and contribute to the knowledge about the accumulation of fatty acids of Sacha Inchi.
All the data were presented in detail, they are robust, statistically sound and the experiments were well controlled.
The conslusions were based on the obtained results and they are well stated.

Additional comments

It is a very interesting work that significantly contribute to the knowlegde.

Reviewer 3 ·

Basic reporting

English used throughout the manuscript is clear, unambiguous, and professional.
I have some suggestions:
1. There are a number of abbreviated forms in the whole manuscript so I suggest inserting a new section to write the full forms of all the abbreviated terms.
2. In line 86, mention some of the major limitations of NGS technology.
3. In lines 218 and 220, the units of data are mismatched. Check it.

Experimental design

The methodology and technology were appropriately implemented to get insights into gene data.

Validity of the findings

It would be great if the authors provide the data for further investigation by other researchers in the field.

Reviewer 4 ·

Basic reporting

The language expression of the manuscript is relatively clear, the analysis is comprehensive and detailed, and the experimental design has some shortcomings, but it still has great reference significance.

Experimental design

No comment.

Validity of the findings

No comment.

Additional comments

1. The seed maturation cycle of about 110 days needs a shorter sampling interval, and the three sampling points are not representative enough to observe the complex changes in the process of seed development.
2.How to use Second-generation transcriptome sequencing data to correct SMRT full-length transcriptome data?
3. The key genes of fatty acid biosynthesis need real-time quantitative preliminary verification.
4. The summary does not need to repeat the results, and it is recommended to further refine.

---

## Round 0.2 · Minor Revisions

Please revise the manuscript as per suggestion by reviewer (especially reviewer 4) and submit.

Reviewer 1 ·

Basic reporting

YES

Experimental design

YES

Validity of the findings

YES

Reviewer 4 ·

Basic reporting

no comment

Experimental design

no comment

Validity of the findings

no comment

Additional comments

This paper has been revised for the questions I raised. But I recommend that a minor revision is warranted.
1. Introduction, the progress of reproductive biology of Plukenetia volubilis should be added.
2. Materials and methods, how to obtain two types of plant materials and how to correct SMRT full-length transcriptome data? please provide the description of the necessary as far as possible.

---

## Round 0.3 · accepted · Accept

Acceptable for publication.